# VaRT: Variational Regression Trees

**Sebastian Salazar Escobedo**
Department of Computer Science
Columbia University
New York, NY 10027
sebastian.salazar@cs.columbia.edu

## Abstract

Decision trees are a well-established tool in machine learning for classification and regression tasks. In this paper, we introduce a novel non-parametric Bayesian model that uses variational inference to approximate a posterior distribution over the space of stochastic decision trees. We evaluate the model's performance on 18 datasets and demonstrate its competitiveness with other state-of-the-art methods in regression tasks. We also explore its application to causal inference problems. We provide a fully vectorized implementation of our algorithm in PyTorch.

## 1 Introduction and Related Work

The use of binary trees for classification and regression problems is widespread in the Machine Learning community (see Breiman [2001], Criminisi [2011]). Chipman et al. [1998] proposed the Bayesian CART method, which offers a fully Bayesian treatment of the problem, aiming to learn a distribution over trees as opposed to a single tree. This innovation led to the development of more sophisticated algorithms, such as Bayesian Additive Regression Trees (BART) (see Chipman et al. [2010]), which has stood the test of time and still shows competitive performance to this day. Both BART and Bayesian CART approximate the posterior distribution over trees through the use of Markov Chain Monte Carlo (MCMC) methods.

In the period following the inception of the Bayesian CART methodology, the field has witnessed significant advancements in approximate Bayesian Inference. This paper explores approximating a posterior distribution over the space of trees via variational inference, a compelling MCMC alternative based on an optimization problem as elaborated in Blei et al. [2017]. Our approach introduces a unique stochastic process over the space of trees, serving as a model prior. Notably, despite the infinite parameter specification of this stochastic process, we show that variational inference remains feasible.

This paper is also related to work on soft decision trees (Kontschieder et al. [2015]), where the goal is to learn a stochastic decision tree of a fixed structure by minimizing some sort of loss function (see Section 2.1). In this paper, we adopt a fully Bayesian perspective to this problem and instead learn a distribution over soft decision trees of varying structure. A fully Bayesian approach has the advantage in that it allows us to naturally quantify the uncertainty of the predictions made by our model.

Overall, our contributions in this paper include a novel stochastic process over the space of trees, a variational inference algorithm for approximating the posterior distribution over trees, and a demonstration of the performance of our model on both regression and causal inference tasks. We believe that this work opens up new avenues for future research in the area of Bayesian Non-parametrics and decision trees.

37th Conference on Neural Information Processing Systems (NeurIPS 2023).

## 2 Model

### 2.1 Stochastic Decision Trees

A stochastic tree, denoted as $T$, is a binary tree where each of its $b-1$ internal nodes is associated with a soft decision rule. This rule indicates the probability of a particular data point taking the left branch of the tree. In this paper, we focus on decision rules of the form $P(\text{datapoint } x \text{ at node } i \text{ takes the left branch}) = \sigma(\beta_i^T X)$. Moreover, we introduce a prior probability measure, denoted as $\Pi(\Theta, T)$, that encompasses the space of trees $T$ (defined by splits and decision rules) and the associated leaf parameters $\Theta$. The generative process for the data is divided into two parts:

1. Prior Distribution:

$$(T, \Theta) \sim \Pi(T, \Theta) \tag{1}$$

Unlike BART or Bayesian CART, our prior distribution is designed differently, incorporating a sampling algorithm that generates a collection of parameters used to fully and uniquely reconstruct a tree. The second part of the model is the likelihood that gives rise to the observed data

2. Likelihood:

$$(y_i | X_i, T, \Theta) \sim p(y_i | x_i, T, \Theta) \tag{2}$$

The likelihood represents the conditional distribution of $y_i$ given $x_i$, $T$, and $\Theta$. In our case, it is usually modeled as a Normal distribution. Further details regarding the likelihood and the overall model will be discussed in subsequent sections.

### 2.2 Likelihood

We associate each of the $b$ leafs of a stochastic tree $T$ with a parameter $\Theta := \{\theta_1, ..., \theta_b\}$. Every parameter $\theta_i \in \Theta$ specifies a density over $\mathcal{Y}$, denoted by $f(y|\theta_i)$.[1] Throughout, we assume that the features $x \in \mathcal{X}$ are fixed, and that the target variables $y \in \mathcal{Y}$ are sampled from a distribution whose parameters stochastically determined by having $x \in \mathcal{X}$ traverse down the tree. In other words, if $x \in \mathcal{X}$ is assigned to leaf $i$, then $y \in \mathcal{Y}$ is sampled from $f(y|\theta_i)$. If a feature $x \in \mathcal{X}$ is assigned to leaf node $i$ with a certain probability, this implies that the likelihood $p(y|X, T, \Theta)$ is a mixture distribution over the densities $f(y|\theta_i)$ and is given by

$$p(y|T, \Theta, x) = \sum_{i \in \text{Leaf}} P(x \text{ takes a path leading to node } i) f(y|\theta_i). \tag{3}$$

### 2.3 A Bayesian non-parametric process prior over the space of stochastic trees $\Pi(\Theta, T)$

We define a prior distribution over a stochastic decision trees by specifying a sampling algorithm. The joint distribution over decision trees $T$ and it's corresponding leaf parameters $\Theta$ is specified through an infinite number of parameters that have a one-to-one correspondence with characteristics of a stochastic decision tree. The parameters required to draw a sample from the joint distribution over $(T, \Theta)$ —which we denote by $\Pi(T, \Theta)$ are as follows [2]

1. $\{\gamma_i\}_{i=1}^{\infty} \in [0, 1]^{\infty}$ specify the probabilities of splitting a node in the tree. In other words, node $i$ in the tree is split with probability $\gamma_i$.

2. $\{\mu_i\}_{i=1}^{\infty}$ are the parameters that define the distribution over the weights specifying the splitting rules.

3. $\{\zeta_i\}_{i=1}^{\infty}$ are the parameters that define the conditional likelihoods over the observations falling in node $i$, provided that it is a leaf.

---

[1] The densities are relative to an appropriate dominating measure $\nu$. In practice, $\nu$ will usually be the Lebesgue or the counting measure.

[2] In this paper, we adopt the following convention for numbering the nodes of a binary tree: (a) We start by labeling the root node as 1 (b) recursively label the left children of node $j$ with $2j$ and the right children of node $j$ with $2j + 1$.

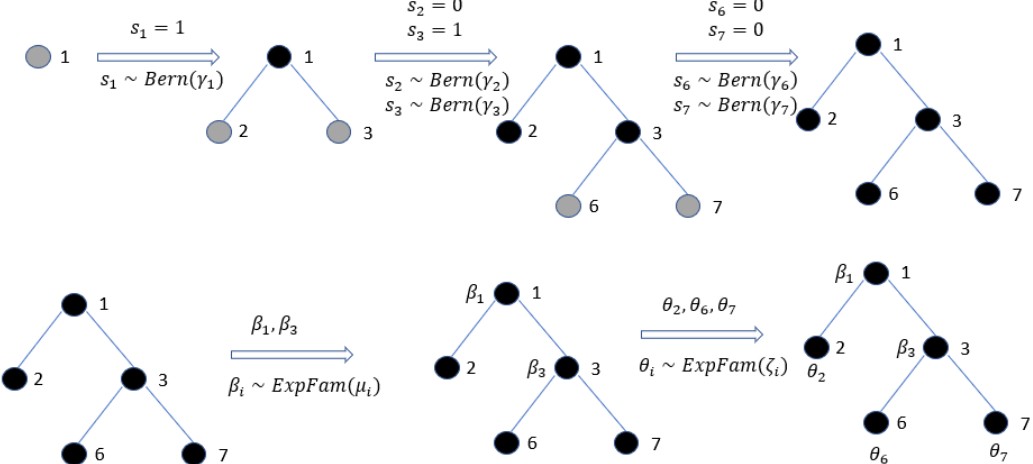

Figure 1: Taking a sample from the generative process

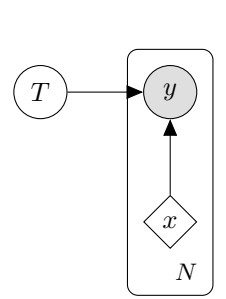

Figure 2: Graphical Model associated to the data generating process.

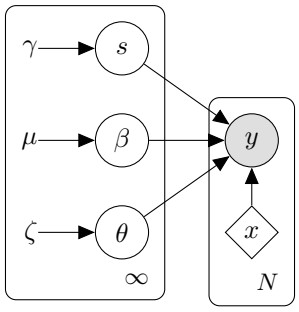

Figure 3: Graphical Model associated to the data generating process. The parameters $s$, $\beta$, and $\theta$ are used to uniquely specify a stochastic decision tree and its parameters.

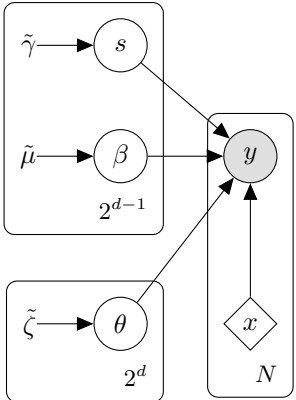

Figure 4: Variational approximation to the data generating process.

Given these parameters, a stochastic tree is sampled as follows:

1. For each node node $i \in \{1, 2, ...\}$

   (a) Sample $s_1, ..., s_k, ... \sim \text{Bernoulli}(\gamma_i)$
   (b) Sample $\beta_1, ..., \beta_k, ... \sim \text{ExpFam}(\mu_i)$
   (c) Sample $\theta_1, ..., \theta_k, ... \sim \text{ExpFam}(\zeta_i)$

Note that the parameters $(s_1, ..., \beta_1, ..., \theta_1, ...)$ suffice to completely specify a binary stochastic tree. We give a concrete example of this generative process in Figure 1 and the graphical model associated to this data-generating process is shown in 3.

## 2.4 A variational family over the space of trees and tree parameters $q(\Theta, T)$

A variational family is obtained by truncating the Bayesian Nonparametric process described in section 2.3 at a fixed tree depth $d$. This truncation is depicted as a graphical model in figure 4. A similar pattern is observed in variational inference for Dirichlet Process Mixtures proposed by Blei and Jordan [2006], where the stick-breaking process of the Dirichlet Process mixture is also

truncated after a fixed number of steps. It is important to note that only the variational distribution is truncated, not the underlying model itself. Furthermore, we would like to emphasize that the choice of exponential family in the variational distribution aligns with those chosen for the prior distribution. To distinguish them from the prior parameters, we label the variational parameters with a tilde (i.e. $\tilde{\gamma}, \tilde{\mu}, \tilde{\zeta}$) on top.

## 2.5 Maximizing the ELBO

We obtain an approximation of the posterior distribution by maximizing a weighted version of the evidence lower bound (ELBO) with respect to the model parameters:[3]

$$\text{ELBO}(q) = \mathbb{E}_q \left( \log p(Y|X, \Theta, T) \right) - \lambda \text{KL}(q(\Theta, T)||p(\Theta, T)) \tag{4}$$
$$\text{ELBO}(q) = \mathbb{E}_q \left( \log p(Y|X, \Theta, T) \right) \tag{5}$$
$$+ \lambda \mathbb{E}_q \left( \log p(\Theta, T) \right) \tag{6}$$
$$- \lambda \mathbb{E}_q \left( \log q(\Theta, T) \right). \tag{7}$$

Where $\lambda$ is a hyperparameter controlling the amount of regularization. The gradients of the variational objective are approximated stochastically with Monte Carlo samples from the variational family. To obtain low-variance estimates of the gradient estimates, we make use of the reparametrization trick (see Kingma and Welling [2022], Ruiz et al. [2016]), where a Softmax-Gumbel approximation is used for the Bernoulli random variables (Jang et al. [2017]). Other approaches for variance-reduction like Rao-Blackwellization are possible but these ideas were not explored in this paper (see Ranganath et al. [2013]). Given a sample from the variational family $(s_1, ..., s_{2^{d-1}}, \beta_1, ..., \beta_{2^{d-1}}, \theta_1, ..., \theta_{2^d})$, we evaluate each of the terms in the ELBO as follows:

$$\log p(y|T, \Theta, x) = \log \left( \sum_{i \in \text{Leaf}} P(x \text{ takes a path leading to node } i) f(y|\theta_i) \right) \tag{8}$$

$$\log p(\Theta, T) = \sum_{i \in \text{Internal Nodes}} \log \left( \gamma_i \right) + \sum_{i \in \text{Internal Nodes}} \log \left( \text{expFam}(\beta_i; \mu_i) \right) \tag{9}$$

$$+ \sum_{i \in \text{Leaves}} \log \left( 1 - \gamma_i \right) + \sum_{i \in \text{Leaves}} \log \left( \text{expFam}(\theta_i; \zeta_i) \right) \tag{10}$$

$$\log q(\Theta, T) = \sum_{i \in \text{Internal Nodes}} \log \left( \tilde{\gamma}_i \right) + \sum_{i \in \text{Internal Nodes}} \log \left( \text{expFam}(\beta_i; \tilde{\mu}_i) \right) \tag{11}$$

$$+ \sum_{i \in \text{Leaves not in last level}} \log \left( 1 - \tilde{\gamma}_i \right) + \sum_{i \in \text{Leaves}} \log \left( \text{expFam}(\theta_i; \tilde{\zeta}_i) \right) \tag{12}$$

In order to make use of reparametrization gradients, each of the three terms in the ELBO must be written in terms of the model parameters (i.e. $\{\gamma, \mu, \zeta, \tilde{\gamma}, \tilde{\mu}, \tilde{\zeta}\}$ ) and a reparametrization for each of the samples must be specified. These calculations are deferred to the next section.

## 3 Variational Regression Trees

In this paper, we explore the use of Variational Regression Trees. We propose using the normal distribution as a natural choice for the splitting and prediction vectors of the prior and variational families, as follows:[4]

$$\beta_i \sim \mathcal{N}(\mu_i, S_i) \tag{13}$$
$$\theta_i \sim \mathcal{N}(\zeta_i, \Psi_i) \tag{14}$$

Where $S_i$ and $\Psi_i$ are diagonal covariance matrices. We also assign a normal distribution to each of the leaves in the tree by setting $f(y|\theta_i)$ to a normal distribution with parameters $\mathcal{N}(\theta_i^T x, \sigma_i^2)$. The normal distribution is a natural choice for regression problems and allows for an easy application of

---

[3]Whenever $\lambda = 1.0$, this is equivalent to minimizing the KL divergence between the variational family and the posterior distribution over the space of trees.

[4]This shows the parameters of the prior, the corresponding parameters for the variational family are denoted with the same letters but with a tilde on top.

the reparametrization trick. For classification, adding a probit link function to the likelihood should result in a tractable variational inference algorithm. We believe including a logit link on the outcome $y$ is also possible, but would require a Poly-Gamma augmentation (see Polson et al. [2013]). We leave all of these modifications to future work (see section 5 for an overview on potential future work).

## 3.1 Learning the Variational Family

The parameters of the variational family are learned by minimizing Monte Carlo approximations of the ELBO with gradient descent. The naive Monte Carlo Estimators for this types of problem are given by $\nabla_\phi \mathbb{E}_{q_\phi(x)}(f(x)) = \mathbb{E}_{q_\phi}(f(x)\nabla_\phi \log q_\phi(x)) \approx \frac{1}{N}\sum_{i=1}^{N} f(x^{(i)})\nabla_\phi \log q_\phi(x^{(i)})$. Although asymptotically consistent, this estimator exhibits extremely high variance, rendering it impractical for the purposes of stochastic optimization. Instead, we make use of a well-known variance-reduction trick through a reparametrization of the samples obtained from the variational family. In particular, we rewrite samples from the variational as a function of an auxiliary noise variable as follows

$$\epsilon \sim p(\epsilon) \tag{15}$$

$$x = g(\phi, \epsilon). \tag{16}$$

The reparametrization of the normal distribution of equations 13 and 14 is straightforward and is given below

$$\epsilon_{rules}, \epsilon_{leaves} \sim_{iid} \mathcal{N}(0, I_d) \tag{17}$$

$$\beta_i = S_i^{1/2}\epsilon_{rules} + \mu_i \tag{18}$$

$$\theta_i = \Psi_i^{1/2}\epsilon_{leaves} + \zeta_i \tag{19}$$

Reparametrizing the Bernoulli random variables requires use of the Softmax-Gumbel trick of Jang et al. [2017]. The key insight relies on the fact that samples from a Bernoulli distribution with success probability $\gamma$ may be obtained as follows

$$v = \text{one\_hot}\left(\arg\max\left(g_1 + \log(\gamma), g_2 + \log(1 - \gamma)\right)\right) \tag{20}$$

$$s = (1, 0)^T v \tag{21}$$

Where $g_1, g_2$ are independent samples from a Gumbel$(0, 1)$ distribution. In practice, we approximate equation 20 using the following limitting representation of the onehot argmax function

$$v = \lim_{\tau \to 0^+} \text{softmax}\left(\frac{g_1 + \log(\gamma)}{\tau}, \frac{g_2 + \log(1 - \gamma)}{\tau}\right). \tag{22}$$

During training, we set $\tau$ to a fixed value and then anneal it towards zero as the training progresses. These reparametrizations allow us to use a single sample from the variational family to approximate the ELBO. To tie everything together and make use of the reparametrization trick, we need to rewrite each of the terms in equations 8 to 12 solely in terms of the sample parameters $(s, \beta, \theta)$ and the distributional parameters $(\gamma, \tilde{\gamma}, \mu, S, \tilde{\mu}, \tilde{S}, \zeta, \Psi, \tilde{\zeta}, \tilde{\Psi})$. This is done by replacing the sums over the leaves and internal nodes with indicator functions, which may be written as products over the node splitting indicator variables $s = (s_1, ..., s_{2^d-1})$. If we denote the internal and leaf node indicator functions for node $i$ by $\psi_i$ and $\kappa_i$ respectively, we may rewrite each of the terms in the ELBO as follows

$$\log p(\Theta, T) = \sum_{i=1}^{b-1} \psi_i \left(\log\left(\gamma_i\right) + \log \mathcal{N}(\beta_i; \mu_i, S_i)\right) \tag{23}$$

$$+ \sum_{i=1}^{2b-1} \kappa_i \left(\log\left(1 - \gamma_i\right) + \log \mathcal{N}(\theta_i; \zeta_i, \Psi_i)\right) \tag{24}$$

$$\log q(\Theta, T) = \sum_{i=1}^{b-1} \psi_i \left(\log\left(\tilde{\gamma}_i\right) + \log \mathcal{N}(\beta_i; \mu_i, S_i)\right) \tag{25}$$

$$+ \sum_{i=1}^{b-1} \kappa_i \log\left(1 - \tilde{\gamma}_i\right) + \sum_{i=1}^{2b-1} \kappa_i \log \mathcal{N}(\theta_i; \zeta_i, \Psi_i) \tag{26}$$

$$\log p(Y|T, \Theta, X) = \log\left(\sum_{i=1}^{2b-1} \kappa_i \prod_{k=1}^{\lfloor \log i \rfloor} \sigma\left((-1)^{\text{pa}_{k-1}(i)} \beta_{\text{pa}_k(i)}^T x\right) f(y|\theta_i)\right) \tag{27}$$

Where $\psi_i$ and $\kappa_i$ are given by

$$\psi_i = s_i \prod_{k=1}^{\lfloor \log i \rfloor} s_{\mathrm{pa}_k(i)} \tag{28}$$

$$\kappa_i = \begin{cases} (1 - s_i) \prod_{k=1}^{\lfloor \log i \rfloor} s_{\mathrm{pa}_k(i)}, i \in \{1, ..., b-1\} \\ \prod_{k=1}^{\lfloor \log i \rfloor} s_{\mathrm{pa}_k(i)}, i \in \{b, ..., 2b-1\} \end{cases} \tag{29}$$

### 3.2  Posterior Predictive Distribution and Bayesian Confidence Intervals

The fully Bayesian treatment of the problem enables us to obtain natural confidence intervals on our predictions, which is a significant advantage of the Bayesian approach compared to other black-box machine learning algorithms. Given a dataset $\mathcal{D} = \{(x_i, y_i)\}_{i=1}^{N}$, confidence intervals for the predictions are obtained by approximating the predictive posterior distribution as follows:

$$p(y'|\mathcal{D}, x') = \int p(y', T, \Theta|\mathcal{D}, x') d\Pi(T, \Theta) \tag{30}$$

$$= \int p(T, \Theta|\mathcal{D}) p(y'|T, \Theta, x') d\Pi(T, \Theta) \tag{31}$$

$$\approx \int q(T, \Theta) p(y'|T, \Theta, x') d\Pi(T, \Theta). \tag{32}$$

Equation 31 shows that the predictive posterior is a mixture of the posterior and the likelihood. Thus, to obtain samples from the predictive posterior distribution, we sample a tree with its parameters $(\Theta, T)$ from the posterior and and then use $(\Theta, T)$ to obtain a sample from the likelihood. Moreover, as we do not have access to the true posterior distribution, we use the learned variational family $q(T, \Theta)$ as an approximation. For prediction purposes, we estimate the posterior predictive mean using the mixture representation of equation 32.

## 4  Experiments

This section presents a comparison of the performance of Variational Regression Trees with three other widely used boosting algorithms: CatBoost, XGBoost, and RandomForest. A comparison with Bayesian Additive Regression Trees (BART, Chipman et al. [2010]) and a Multilayer Perceptron (MLP, LeCun et al. [2015]) is deferred to the Appendix. We conducted experiments on 18 distinct Datasets from the UCI Machine Learning Repository to benchmark our algorithm. In addition, we explored the use of VaRT for estimating of conditional treatment effects in causal inference problems, motivated by the fact that BART is a popular algorithm for such problems.

### 4.1  Performance on Regression Datasets

The results of our regression experiments are shown in Table 2. We compare the performance of VaRT to three popular Boosting Algorithms: CatBoost, XGBoost, and RandomForest. We compared an ensemble of fifty trees for each algorithm to single VaRT trees of depths 3,5,7, and 10. A comparison with BART and Multilayer Perceptrons is given in the Appendix.

In these experiments, VaRT was trained using gradient descent paired with a Clipped Adam optimizer in PyTorch (Paszke et al. [2019], Bingham et al. [2018]). A single VaRT tree for depths $3, 5, 7$, and $10$ were trained for each dataset. We report the average RMSE values for VaRT across all depths to provide a thorough representation of VaRT's performance across various configurations (Table A). The regularization parameters for all runs was set to $10^{-3}$ and no hyperparameter tuning was performed to make a fair comparison on the off-the-shelf performance of each of the algorithms. In practice, both the regularization and truncation parameters could be chosen through cross-validation. Early stopping by evaluating the RMSE on a training or validation set everytime a fixed number of epochs are completed should also increase performance.

Compared to other tree-based Bayesian algorithms, such as BART, VaRT offers distinct advantages that enhance its practicality. Notably, VaRT's framework is inherently parallelizable, a feature that sets it apart from MCMC-based methods like BART which are inherently sequential and lack this

| Dataset | CatBoost | RandomForest | XGBoost | VaRT3 | VaRT5 | VaRT7 | VaRT10 |
|---|---|---|---|---|---|---|---|
| autompg | 0.121 | 0.115 | 0.113 | 0.116 | 0.116 | 0.116 | **0.109** |
| concreteslump | 0.209 | 0.237 | 0.252 | **0.009** | **0.009** | 0.028 | 0.047 |
| forest | 0.396 | 0.403 | 0.403 | 0.392 | 0.394 | **0.387** | **0.387** |
| solar | **0.086** | **0.086** | **0.086** | 0.114 | 0.092 | 0.094 | 0.093 |
| stock | 0.074 | 0.067 | 0.072 | 0.065 | 0.068 | 0.065 | **0.059** |
| yacht | 0.036 | 0.026 | **0.018** | 0.041 | 0.035 | 0.035 | 0.046 |
| airfoil | 0.073 | 0.062 | **0.061** | 0.151 | 0.133 | 0.131 | 0.131 |
| autos | 0.092 | **0.072** | 0.076 | 0.121 | 0.146 | 0.134 | 0.140 |
| breastcancer | 0.403 | 0.362 | 0.398 | 0.390 | 0.476 | 0.380 | **0.360** |
| concrete | 0.095 | 0.093 | **0.081** | 0.149 | 0.141 | 0.158 | 0.148 |
| fertility | 0.402 | 0.375 | **0.358** | 0.481 | 0.454 | 0.456 | 0.387 |
| housing | 0.103 | 0.118 | 0.120 | 0.101 | **0.097** | 0.098 | 0.101 |
| machine | **0.089** | 0.112 | 0.097 | 0.110 | 0.123 | 0.124 | 0.138 |
| pendulum | 0.061 | 0.068 | 0.074 | 0.077 | 0.064 | **0.056** | 0.058 |
| servo | 0.066 | 0.062 | **0.050** | 0.096 | 0.077 | 0.078 | 0.109 |
| skillcraft | 0.118 | 0.116 | 0.120 | **0.114** | **0.114** | **0.114** | **0.114** |
| sml | 0.028 | **0.018** | 0.020 | 0.063 | 0.056 | 0.049 | 0.051 |
| wine | 0.108 | 0.103 | **0.096** | 0.122 | 0.119 | 0.117 | 0.121 |

Table A: Comparison of RMSE values for different boosting algorithms and VaRT. The RMSE values correspond to the average RMSE on the test sets of a random $90/10\%$ train-test split of the data over ten runs. Bold values indicate the lowest RMSE. Each of the ensemble methods were trained using fifty trees. VaRT's performance is shown for trees of depth $3, 5, 7$, and $10$.

| | VaRT | CatBoost | RandomForest | XGBoost |
|---|---|---|---|---|
| tamielectric (45781 datapoints) | 0.508 | **0.503** | 0.545 | 0.504 |
| kin40k (40000 datapoints) | 0.099 | 0.100 | **0.097** | 0.110 |

Table B: RMSE values for different boosting algorithms and VaRT on datasets containing more than $40,000$ datapoints. The RMSE values correspond to the average RMSE on the test sets of a random $90/10\%$ train-test split of the data over six runs. Bold values indicate the lowest RMSE. Each of the ensemble methods were trained using fifty trees, and a variational family of depth $5$ was used for VaRT.

parallelization potential. This parallelizability not only accelerates the model training process but also empowers efficient exploration of large datasets, making VaRT particularly well-suited for modern computation demands. Furthermore, VaRT leverages gradient-based optimization techniques, making it compatible with widely used automatic differentiation engines such as PyTorch. This was validated by our experiments on datasets with over 40,000 datapoints, where training took less than 3 minutes.

We conducted all experiments on an ASUS Zephyrus G14 laptop with an RTX 2060 Max-Q GPU (6 GB VRAM), a 4900HS AMD CPU, 40 GB of RAM (although RAM usage was kept below 6 GB). Each experiment took roughly $\sim$ 1-3 minutes per dataset. Code with the random seeds needed to replicate the results of this section are provided with the supplementary material.

## 4.2 Causal Inference case-study

We investigate the performance of Variationial Regression Trees on several causal inference problems using synthetic and semi-synthetic data. Our experiments are similar to those conducted by Hill [2011]. To introduce causal inference, we consider the potential outcomes that would have been observed for an individual under treatment and no treatment. We denote these quantities as $Y(1)$ and $Y(0)$, respectively. The observed outcome is denoted by $Y$ and is related to the potential outcomes through the formula $Y = AY(1) + (1 - A)Y(0)$, where $A$ is a dichotomous variable indicating whether a particular individual in the population recieved treatment. Our goal is to estimate different quantities that quantify the effect of a treatment for various subpopulations.

In section 4.2.1, we focus on the conditional average treatment effect (CATE), which can be identified through the following formula under standard causal inference assumptions, including overlap, conditional exchangeability, and consistency:

$$\mathbb{E}(Y(1) - Y(0)|X = x) = \mathbb{E}(Y|A = 1, X = x) - \mathbb{E}(Y|A = 0, X = x). \tag{33}$$

In section 4.2.2, our goal is to estimate the Average Treatment Effect (ATE), which under standard causal inference assumptions can be given by:

$$\mathbb{E}(Y(1) - Y(0)) = \mathbb{E}(\mathbb{E}(Y|A = 1, X) - \mathbb{E}(Y|A = 0, X)). \tag{34}$$

While we cannot provide a comprehensive treatment of causal inference in this paper, we refer interested readers to Pearl et al. [2016], M. [2023].

### 4.2.1 Experiment on Toy Data

We begin by generating two toy synthetic datasets using the following generative process

$$A_i \sim \text{Bernoulli}(0.5) \tag{35}$$

$$X_i|A_i = a_i \sim a_i \mathcal{N}(\mu_1, 10^2) + (1 - a_i)\mathcal{N}(\mu_2, 10^2) \tag{36}$$

$$Y_i|A_i = a_i, X_i = x_i \sim a_i \mathcal{N}(90 + e^{0.06x_i}, 1) + (1 - a_i)\mathcal{N}(72 + 3\sqrt{x_i}, 1) \tag{37}$$

In the "partially overlapping" setting (Figure 6), the means of the normal distributions used to generate the data are $(\mu_1, \mu_2) = (40, 20)$, as in Hill [2011]. In contrast, in the "overlapping" setting, we set $(\mu_1, \mu_2) = (30, 40)$. The figures illustrating these hypothetical scenarios are Figures 5 and 6. In these figures, the red curve (top) represents the conditional expectation of the potential outcome under treatment $Y(1)$, while the blue curve (bottom) represents the conditional expectation of the potential outcome under no treatment $Y(0)$. To showcase the posterior predictive mean of VaRT for each potential outcome, we use dashed lines alongside 80% Bayesian confidence bands. These bands grow wider in areas where the density of the covariates is low and shrink in areas where the density of the covariates is high. This type of qualitative behavior may be attributed to the Bayesian prior, which acts as a type of regularization on the fit of the model and helps us avoid overfitting.

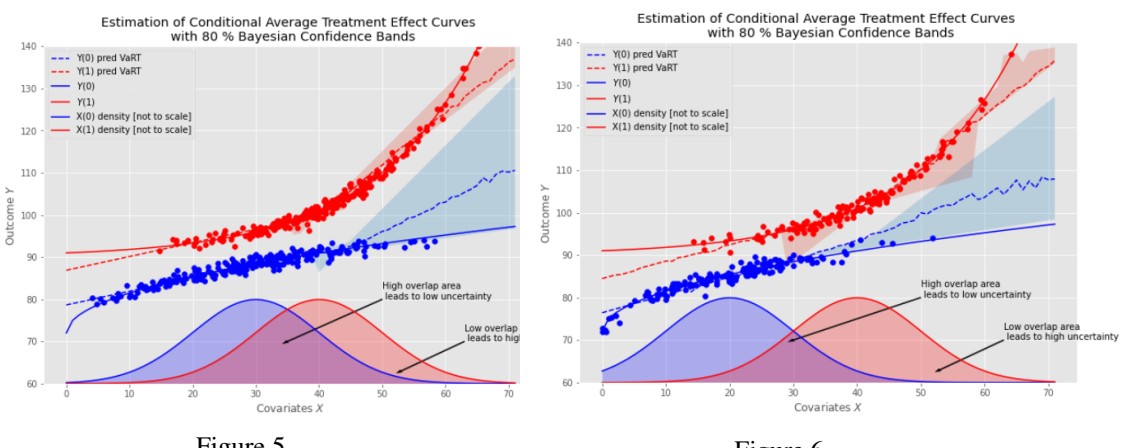

Figure 5        Figure 6

### 4.2.2 Experiments on the Infant Health and Development Program (IHDP) Dataset

We use the IHDP dataset to simulate the two response surfaces of Hill [2011] and then compare the performance VaRT relative to BART when estimating the average causal effect. The first response is linear and it is generated as follows

$$\pi \sim \text{Cat}((0.5, 0.2, 0.15, 0.1, 0.05)) \tag{38}$$

$$\beta_i \sim \pi \cdot (0, 1, 2, 3, 4) \quad i \in \{1, ..., 25\} \tag{39}$$

$$Y(0) \sim \mathcal{N}(X^T\beta, 1) \tag{40}$$

$$Y(1) \sim \mathcal{N}(X^T\beta + 4, 1) \tag{41}$$

| Response Surface | VaRT | BART | Ground Truth |
|---|---|---|---|
| Linear | **3.86 ± 0.12** | 3.6 ± 0.6 | 4.00 |
| Non-Linear | **3.80 ± 0.19** | 2.9 ±0.8 | 4.00 |

Table 1: Estimation of the ATE for the response surfaces described by equations 38 through 45. The results report the mean ACE along with the standard deviation as the error across 5 runs. All of the VaRT trees were trained using a hyperparameter of $\lambda = 1$. The algorithm whose mean values were closer to the Grount Truth are shown in bold.

The second response surface is highly non-linear and it is generated as follows

$$\pi \sim \text{Cat}((0.6, 0.1, 0.1, 0.1, 0.1)) \tag{42}$$

$$\beta_i \sim \pi \cdot (0, .1, .2, .3, .4) \quad i \in \{1, ..., 25\} \tag{43}$$

$$Y(0) \sim \mathcal{N}(\exp(X^T \beta + 1/2), 1) \tag{44}$$

$$Y(1) \sim \mathcal{N}(X^T \beta - \omega, 1) \tag{45}$$

Here, $\omega$ is chosen so that the CATE is equal to 4 for every $X$ (hence the ATE is also 4). The results of these two experiments are reported in table 1 and suggest that VaRT clearly outperforms BART in the task of Causal Effect Estimation on both response surfaces.

## 5 Conclusion and Future Work

In this paper, we presented a novel Bayesian non-parametric process over the space of stochastic decision trees and derived a variational approximation to it. We also proposed a variational inference algorithm to approximate the Bayesian posterior distribution, which we evaluated on both regression and causal inference tasks. Our experiments showed that our model's performance is competitive with other state-of-the-art methods in machine learning.

Looking ahead, future research could explore extensions of the non-parametric Bayesian prior over trees to model forests, similar to what we observe in BART. Additionally, other potential lines of work could include modifying the prior to introduce dependencies among the tree parameters or extending the model to classification.

Overall, the findings of this research provide a promising avenue for future research in Bayesian non-parametrics and decision trees. We hope that this work will inspire other researchers to build upon our model and improve its versatility and performance in various machine learning applications.

# 6 Aknowledgements

This research greatly benefited from discussions with Aaron Schein, Nihar Mauskar, and Ansaf Salleb-Aouissi. Their support was invaluable. Additionally, I thank David Blei for his enlightening seminar on Applied Causal Inference, which inspired this paper. Special thanks to Emily and Michael Teti for their unwavering support during the rebuttal process.

This research was partially supported by NIH/NLM (www.nlm.nih.gov) grant R01LM013327.

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

## A   Comparison with BART and MLP

| | VaRT | | | BART | MLP |
|---|---|---|---|---|---|
| | Median | top 10% | bottom 10% | 50 trees | (100,50) neurons |
| autompg (N=392, d=7) | 0.0751±0.0031 | **0.0708±0.0031** | 0.085±0.005 | **0.0710±0.0020**\* | 0.088±0.013 |
| concreteslump (N=103, d=7) | **0.049±0.023** | 0.034±0.017 | 0.069±0.031 | 0.149±0.011 | 0.16±0.10 |
| forest (N=517, d=12) | 0.205±0.005 | **0.194±0.005** | 0.219±0.008 | **0.1945±0.0026**\* | 0.1977±0.0026 |
| solar (N=1066, d=10) | 0.114±0.006 | 0.108±0.004 | 0.124±0.013 | **0.1038±0.0004** | 0.1044±0.0009 |
| stock (N=536, d=11) | **0.0351±0.0014** | 0.0336±0.0009 | 0.040±0.005 | 0.051±0.004 | 0.060±0.017 |
| yacht (N=308, d=6) | **0.059±0.007** | 0.053±0.007 | 0.069±0.012 | 0.0684±0.0018 | 0.080±0.004 |
| airfoil (N=1503, d=5) | **0.0828±0.0026** | 0.0776±0.0020 | 0.0894±0.0029 | 0.087±0.005 | 0.092±0.008 |
| autos (N=159, d=25) | 0.107±0.007 | 0.0948±0.0031 | 0.121±0.012 | **0.075±0.006** | 0.093±0.004 |
| breastcancer (N=194, d=33) | 0.244±0.017 | 0.220±0.014 | 0.273±0.023 | 0.219±0.008 | **0.205±0.010** |
| concrete (N=1030, d=8) | 0.0837±0.0013 | **0.0786±0.0020** | 0.0922±0.0025 | **0.0768±0.0012**\* | 0.085±0.005 |
| fertility (N=100, d=9) | 0.197±0.023 | 0.180±0.018 | 0.214±0.027 | 0.160±0.005 | **0.158±0.013** |
| housing (N=506, d=13) | 0.074±0.005 | 0.068±0.004 | 0.082±0.006 | **0.069±0.006**\* | 0.077±0.006 |
| machine (N=209, d=7) | 0.093±0.007 | 0.083±0.009 | 0.104±0.009 | **0.0695±0.0013** | 0.14±0.04 |
| pendulum (N=630, d=9) | **0.060±0.006** | 0.054±0.006 | 0.073±0.009 | 0.0697±0.0016 | 0.072±0.006 |
| servo (N=167, d=4) | 0.105±0.013 | 0.090±0.012 | 0.127±0.015 | **0.0827±0.0034** | 0.110±0.006 |
| skillcraft (N=3338, d=19) | **0.0692±0.0006** | 0.0676±0.0007 | 0.0738±0.0031 | 0.0699±0.0007 | 0.0692±0.0025 |
| sml (N=4137, d=26) | **0.0356±0.0029** | 0.0342±0.0022 | 0.041±0.004 | 0.0416±0.0009 | 0.0384±0.0033 |
| wine (N=1599, d=11) | **0.0746±0.0022** | 0.0713±0.0020 | 0.0806±0.0034 | 0.0947±0.0031 | 0.0751±0.0015 |
| **TOTAL** | 8 | - | - | 8 | 2 |

Table 2: Root Mean-Squared Error (RMSE) for VaRT, BART, and a Feedforward Neural Network on 18 regression datasets from the UCI Machine Learning Repository. For VaRT we show the $10^{th}$ and $90^{th}$ Bayesian quantiles along with the standard deviation reported as the error across 5 runs. For the purposes of comparison, we use the median RMSE of VaRT to determine whether it outperformed the other algorithms. The RMSE of the best performing algorithm is shown in bold.
\*: Indicates that the RMSE among the top $10\%$ draws of VaRT's posterior predictive distribution beats (or overlaps) with the RMSE of the corresponding algorithm.

BART was trained using fifty trees and the off-the-shelf prior parameters of Chipman et al. [2010]. The Markov Chain of BART was given a burn-in period of 200 iterations (i.e. the default choice). Finally, the Feed-forward Neural Network (MLP) had two hidden layers with 100 and 50 neurons respectively,, and was trained using an Adam optimizer.

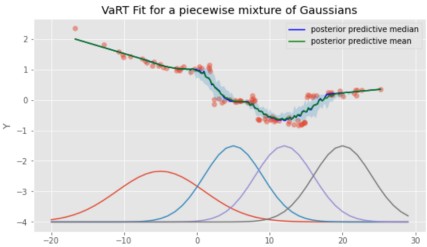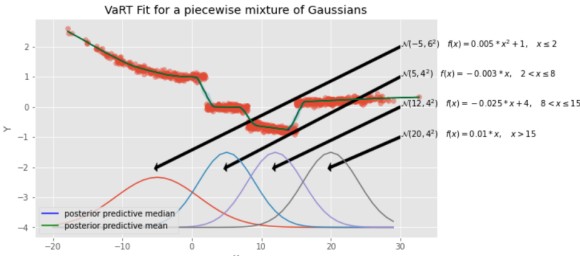

Figure A: Experiments on toy data generated from a piecewise continuous function. The data was generated from a uniform mixture of Gaussians centered on the midpoint of each interval. Analysis was conducted on 100 (left) and 1000 (right) datapoints. Qualitatively it seems like the posterior concentrates around the "true" clusters as we collect more data.

