# OpenReview forum: "VaRT: Variational Regression Trees"
_NeurIPS.cc/2023/Conference — NeurIPS 2023 spotlight_

### Official Review · Reviewer_2ow5 · 2023-07-06

**Soundness:** 3 good
**Presentation:** 2 fair
**Contribution:** 2 fair
**Rating:** 6
**Confidence:** 4

**Summary:**

The authors propose a non-parametric Bayesian model for Regression trees. Variational inference is used approximate the posterior distribution. Extensive experiments are conducted to show the superiors of the method.

**Strengths:**

The authors proposed a non-parametric method to learn regression trees. The paper is well-written and easy to follow. The experimental results on 18 datasets are provided.

**Weaknesses:**

1, The basic idea is too simple. The authors proposed several prior distributions for the variables in tree. Then the parameters in the prior distributions are optimized using SGD with some reparametrization tricks. The priors as well as the reparametrization tricks have been proposed already. So the contribution of this paper is limit.
2, No introduction is provided regarding the 18 datasets. No discussion is provided for the prediction results. The highlighted RMSEs in table 1 are not always the lowest.
3, The datasets used in 4.1 is too small. It is interesting to show the performance on large scale dataset. The computational complexity of the proposed method seems to be high.


**Questions:**

1, what is function g(\phi,\epsilon) in equation (16)?
2, what is the symbol \beta_{pa_k} in equation (27)?
3, Are the RMSEs reported in table 1 of VaRT obtained using a single Tree? Please provides some discussions to explain why VaRT with a single Tree outperforms the BART with 50 Trees. It is interesting to provide the prediction results of  CART and random forest.


**Limitations:**

When compared with CART, the proposed method seems time-consumming

---

> ### Author Rebuttal · Authors · 2023-08-08
>
> Firstly, we extend our gratitude for your assessment of our paper. Your feedback has been instrumental in shaping our work and we value the insights you've shared. We acknowledge your observation about the apparent simplicity of our approach, which centers around non-parametric Bayesian models for regression trees. While the foundational concept of using tree-based models for regression might seem straightforward, we wish to emphasize that the novelty lies in the intricate interplay of the proposed prior distributions, the design of the variational family, and the utilization of these elements within the broader context of Variational Inference to approximate the posterior distribution. We believe that this unique amalgamation contributes to the field by providing a methodologically distinct approach to tackling the challenges associated with Bayesian regression tree modeling.
>
> We firmly believe that this innovation enriches the field by providing an unparalleled framework for addressing the intricacies associated with regression tree modeling in the context of Bayesian models. As of now, we are not aware of any prior research that encompasses the distinctive amalgamation of techniques featured in our method. Given your expertise in the field, we would greatly appreciate any guidance you can provide in identifying references that might shed light on parallel developments.
>
> In your review, you noted the application of a reparametrization trick that has been introduced elsewhere in the literature. We acknowledge this point, and we do not claim originality in this aspect. In fact, we explicitly mention its prior introduction in our manuscript (lines 94-96). Our emphasis was on demonstrating how this established technique fits seamlessly into our broader framework.
>
> We also acknowledge your concerns about the choice of datasets and the absence of discussions regarding prediction results in our paper. We hope to have addressed some of these concerns in the "global" rebuttal and remain open to answering any questions you may have. Regarding the prediction results, our intention was to provide transparent evaluation. The notation in Table 1, indicated by "∗," aimed to highlight instances where VaRT's posterior predictive distribution excelled or overlapped with other algorithms. This approach is aligned with our commitment to unbiased reporting.
>
> In response to your suggestion regarding the dataset sizes employed in our experiments, we took your advice into consideration and extended our evaluation to include datasets comprising more than 40,000 data points. Notably, our method demonstrated seamless scalability to handle these larger datasets, owing to its inherent parallelizability. The computational efficiency of our algorithm was evident, with fitting times spanning a range of one to three minutes. It's worth noting the significant contrast with Bayesian MCMC methods like BART, which are constrained by their sequential nature and lack support for parallelization. While we acknowledge the existence of rapid frequentist boosting algorithms such as XGBoost and CatBoost, we recognize the potential for further exploration in this direction. We are excited about the prospect of future advancements in this area and acknowledge that the initial implementation of our algorithm might not outpace these established algorithms in terms of speed.
>
> Regarding your specific questions, the function $g(\phi,\epsilon)$ is the function utilized in the reparametrization trick. While we adopted this notation from (Kingman and Welling, 22'), we regrettably omitted its explicit definition in our manuscript. We appreciate your attention to this detail and will rectify this oversight by including a dedicated line that provides a clear definition of this function within our paper.
>
> Furthermore, we understand that the notation $\beta_{pa_{k}(i)}$ might have caused confusion. To clarify, this notation represents the $\beta$ coefficient associated with the $k^{th}$ parent of node $i$ in the tree. We recognize the potential for ambiguity and will address this concern by adding a clarification in our paper to ensure proper understanding.
>
> Regarding the observation that a single tree occasionally outperforms an ensemble of trees, we recognize that this might appear unexpected. The distinction lies in our approach's capacity to endow individual trees with the ability to capture intricate data relationships. This capability is rooted in our algorithm's inherent adaptability, facilitated by more complex splitting and prediction rules. Moreover, as we learn a distribution over the space of trees, each posterior sample can yield a different prediction, resulting in an ensemble-like effect. On average, this allows our model to encapsulate inherently intricate functions. This nuanced aspect likely contributes to instances where our singular VaRT posterior distribution over trees showcases superior performance compared to other boosting methods reliant on ensembles of trees. We acknowledge the importance of accentuating this distinctive attribute within our paper to provide a comprehensive context for the observed outcomes.
>
> We want to highlight that we really value your critical insights and are eager to address any concerns you may have to ensure the accuracy and quality of our work. We believe that your perspective will greatly contribute to enhancing our manuscript, and we hope that this clarifying response provides a more accurate portrayal of our contributions and intentions. Thank you once again for your thorough review.
>
> Respectfully,
>
> The Authors.

---

> > ### Comment · Reviewer_2ow5 · 2023-08-15
> >
> > The additive numerical results is quite amazing to me, where a single tree of depth 3 provides competitive results with ensemble methods on about 9 datasets. I agree that the used piece-wise linear regression tree is more expressive than the tree used in RF and GBDT, but the results is still quite amazing. [A1] use the same architecture, but requires multiple trees to achieve similar AUC to cat boost/xgb/lightGBM. Besides, the authors claim that each posterior sample of VaRT can yield a different prediction, resulting in an ensemble-like effect. Can you provide more evidence to support this claim?
> >
> > One of main drawback of Bayesian methods is their higher computational complexity. The results about the runtime of VaRT on large dataset is surprising, the authors claims that this is attributed to their vectorized implementation. So if this vectorized implementation benefit from their specific prior design? If so, I think this is a good work.
> >
> > [A1] Shi, Yu, Jian Li, and Zhize Li. "Gradient boosting with piece-wise linear regression trees." *Proceedings of the 28th International Joint Conference on Artificial Intelligence*. 2019.

---

> > > ### Author Response · Authors · 2023-08-16
> > > **Clarifications on Rebuttal**
> > >
> > > We want to thank you for your detailed response and engagement with the review process. You brought to our attention a couple of very important points that helped us improve the quality of our work and highlight our contributions.
> > >
> > > The reference that you are pointing us to is both interesting and relevant to our work. It is reassuring to see that piecewise linear trees offer an empirical advantage over regular decision trees. To showcase the ensemble-like effect of our method, we decided to sample several trees from the posterior distribution of the toy data of Figure A (left) and look at their topology. We leave four samples from the posterior distribution below
> > >
> > > ```
> > >         _____1_
> > >        /       \
> > >   _____2__     3
> > >  /        \   / \
> > >  4__      5_  6 7
> > > /   \    /  \
> > > 8   9_  10 11
> > >    /  \
> > >   18 19
> > >
> > >  1_________
> > > /          \
> > > 2        __3
> > >         /   \
> > >       __6_  7
> > >      /    \
> > >     12_  13
> > >    /   \
> > >   24  25
> > >
> > >   _1_________
> > >  /           \
> > >  2         __3
> > > / \       /   \
> > > 4 5     __6_  7
> > >        /    \
> > >       12_  13
> > >      /   \
> > >     24  25
> > >
> > >  1_________
> > > /          \
> > > 2        __3__
> > >         /     \
> > >       __6_    7___
> > >      /    \  /    \
> > >     12_  13 14   15_
> > >    /   \        /   \
> > >   24  25       30  31
> > >
> > >     _________1
> > >    /          \
> > >   _2______    3
> > >  /        \
> > >  4      __5_
> > > / \    /    \
> > > 8 9   10_  11
> > >      /   \
> > >     20  21
> > > ```
> > >
> > > We hypothesize that the variation in tree topology, coupled with the stochasticity of node and leaf parameters, contributes to the ensemble-like effect; we also believe this brings some level of smoothness to the posterior predictive mean. This insight is valuable and merits a discussion in the main body of our paper. We intend to incorporate the reference you've shared, alongside a discussion of this ensemble-like phenomenon. Your guidance in this matter is greatly appreciated.
> > >
> > > Regarding your second point, we acknowledge your accurate observation. A core advantage of our prior design lies in its compatibility with vectorized computations. Parallelizability is facilitated by the fact that that both tree sampling a tree and traversing a data point down to a leaf are both the result of a collection of independent computations, thus enabling efficient execution on GPU architectures. This parallelization attribute underscores the rationale behind our utilization of Variational Inference, and offers a significant departure from other MCMC-based Bayesian methods. We concede that our initial manuscript may not have highlighted this aspect of our method adequately and remain committed to revising our paper to emphasize this pivotal contribution.
> > >
> > > We believed that your perspectives have greatly contributed to enhancing the quality of our manuscript. Once again, we thank you for your review and expertise and remain open to addressing any other questions you may have.
> > >
> > > Best,
> > >
> > > The Authors

---

> > > > ### Comment · Reviewer_2ow5 · 2023-08-17
> > > >
> > > > Thank you for your responses. I'll increase my score to weak accept.

---

### Official Review · Reviewer_Ya4w · 2023-07-07

**Soundness:** 4 excellent
**Presentation:** 4 excellent
**Contribution:** 4 excellent
**Rating:** 8
**Confidence:** 5

**Summary:**

This paper introduces a non-parametric Bayesian model taking the form of a stochastic decision tree and proposes to approximate the posterior distribution with variational inference. The prior involves three parts, a prior on the tree structure, a prior on the probabilistic splitting criteria, and a prior on the conditional distribution at each leaf. To conduct variational inference, reparameterization tricks of normal and binary variables are used. Prediction can be made based on the posterior predictive distribution and inference can be made with Bayesian credible intervals. Numerical experiments are supportive, including a comparison with benchmarks on prediction and application to toy datasets as well as real datasets.

**Strengths:**

1. The paper is clear and well written.
2. The proposed method is strong in originality.
3. Numerical experiments are supportive.
4. It addresses the long-lasting problem that Bayesian tree methods (Bayesian CART or BART) are infeasible for larger datasets and their MCMC chains may never really converge. The combination of stochastic decision trees with variational inference suggests a promising direction in handling this problem.

**Weaknesses:**

There doesn't appear to be any major weaknesses to me. The following are some minor ones that could be improved:
1. For Section 4.1, since the task is purely predictive, it would be better to compare with aggregated decision trees or random forest methods as well. Or alternatively, if the comparison is restricted purely to Bayesian models, a Bayesian neural network should be used with parameters inferred either with variational inference or HMC sampler.
2. It would be nice to compare the posterior distribution approximated by variational inference with the true posterior distribution. For instance, with low dimensional covariates and response, a data augmented Gibbs sampler (with binary latent variables at each splitting point) along with reversible jumps could also be designed and used to sample from the true posterior distribution.
3. There seems to be too much parameters introduced with gamma, mu, zeta. Some structural formulations such as a cumulative shrinkage prior with just a few prior parameters might be better.

Of course, all of these would require lots of additional effort, so as a first paper in this direction the lack of them is not a problem.

**Questions:**

1. Since the data generating model could be viewed as a mixture model, the posterior distribution of the tree also gives a posterior distribution over the clusters of data. Does this posterior distribution over the clusters concentrate around the true clusters when the true data generating model is a mixture model?

2. As mentioned in the paper, the variational regression tree can be generalized to variational classification tree with a binary response variable. How does it generalize to classification of response variables with more than two levels? On a side note, for doing variational inference instead of Gibbs sampling, why would Polya-Gamma augmentation still be needed?

3. The soft splitting criterion is based on a logistic regression of the covariates, so the splitted regions is no longer rectangular. While this may be easier for implementation, how does it compare to the case where the soft splitting criterion are all restricted to be parallel to the axis (a hierarchical prior on beta could do this)? Also, for the obtained posterior samples, does the splitting critria at different nodes tend to be more parallel or more orthogonal?

**Limitations:**

Not much, see Weaknesses and Questions.

---

> ### Author Rebuttal · Authors · 2023-08-08
>
> We sincerely appreciate your exceptional review of our manuscript. Your insightful feedback has significantly impacted our research, and we are grateful for your valuable insights. Your evaluation not only validates our efforts but also provides us with invaluable perspectives that will undoubtedly shape the trajectory of our paper and future work, regardless of the review outcome.
>
> Your observation regarding the comparison with other frequentist methods is well-taken. To address this concern, we conducted additional experiments comparing the performance of VaRT against established ensemble methods like Random Forests, XGBoost, and CatBoost. Furthermore, we expanded our experiments to include two datasets, each containing over 40,000 points, to address scalability concerns raised by other reviewers. We are excited to report that our method exhibited seamless scalability and comparable performance to other boosting algorithms (see table A). As you rightfully pointed out, we believe methods based on Variational Methods offer a distinctive advantage due to their inherent parallelizability. This is in stark contrast to other Bayesian methods based on Markov Chain Monte Carlo.
>
> Your suggestion regarding exploring a data augmented Gibbs sampler for posterior distribution comparison is insightful. We are currently investigating similar analyses in the literature to gain a better understanding of this approach. If you have any specific references you could suggest, it would greatly aid our exploration.
>
> We were also particularly intrigued by your mention of cumulative shrinkage priors. Building on this, we plan to explore the application of cumulative shrinkage priors and their implications for our model in future work. We would greatly appreciate your suggestions on relevant references in this regard! (were you thinking of hierarchical priors, like hourshoe priors, on the splitting/leaf nodes to induce sparsity?)
>
> Addressing your specific questions, we conducted experiments on toy data generated from a piecewise continuous function. The data was generated from a uniform mixture of Gaussians centered on the midpoint of each interval. Our results suggest that VaRT correctly "routes" each point to the appropriate node with high probability, as evidenced by our analysis on 100 and 1000 datapoints. This suggests that the posterior concentrates around the "true" clusters.
>
> Regarding the use of Pólya-Gamma augmentation, while it may not be strictly necessary, we have encountered cases where Pólya-Gamma random variables have been integrated into Variational Inference schemes to get closed-form updates of the natural gradients. Notably, the paper "Efficient Gaussian Process Classification Using Pólya-Gamma Data Augmentation" by Wenzel et al. discusses a similar application. However, we now recognize that their use is not strictly necessary and will explicitly note this in our paper and acknowledge your insight.
>
> For generalizing our method to more than two response variables, we propose incorporating a matrix β ∈ ℝ^(c × d), attaching a categorical distribution to each leaf nodes as follows $\text{Categorical}(\text{SoftMax}(βX))$ and employ discrete reparameterization techniques to fit variational family.
>
> In response to your question about hierarchical priors and orthogonal splitting criteria, we believe hierarchical priors, such as a horseshoe priors on the variances of β coefficients, could indeed induce sparsity and enhance model interpretability as it would bias the model towards axis-alignted splits. We believe this is an exciting area for future work! Currently our method does not have any particular bias for this and most of the splits end up being non-axis-aligned.
>
> Thank you once again for your detailed review! Your feedback has undoubtedly enriched our understanding of our work. Regardless of the review outcome, we intend to acknowledge your valuable contributions in the acknowledgments section of our paper. We'll remain available to answer any other questions you may have during the discussion period.
>
> Sincerely,
>
> The Authors

---

> > ### Comment · Reviewer_Ya4w · 2023-08-21
> >
> > Thank you for the further explanations and responses to several questions. I will keep the score as is but I now have a higher confidence for the score.

---

### Official Review · Reviewer_MwKc · 2023-07-07

**Soundness:** 3 good
**Presentation:** 2 fair
**Contribution:** 3 good
**Rating:** 5
**Confidence:** 4

**Summary:**

The paper develops a new generating process for soft decision trees. The proposed process is then adopted as a prior model in a Bayesian nonparametric regression setting, and a novel variational inference algorithm is developed using the truncated version of the tree generating process as the variational distribution. Experiments suggest that the proposed method has comparable performance with BART in real data sets and causal inference applications.

**Strengths:**

* A good addition to the literature of Bayesian soft decision tree models.
* The soft decision trees generating process and the variational inference algorithm is technically sound and novel.
* Competitive performance with BART.

**Weaknesses:**

* Some important baselines are missing. The proposed method is not compared with other soft decision tree based models such as soft BART (Linero and Yang, 2018; Linero, 2022) and the frequentist version of soft decision trees.
* Experiments are only based on some default hyperparameters for both VaRT and the baselines; hyperparameter sensitivity is not investigated.
* The paper can benefit from careful proofreading. For instance, a few notations are used without any definition. Formatting and presentation of the paper can also be improved.

Reference:

[1] Linero, A.R. and Yang, Y., 2018. Bayesian regression tree ensembles that adapt to smoothness and sparsity. Journal of the Royal Statistical Society Series B: Statistical Methodology, 80(5), pp.1087-1110.

[2] Linero A.R. 2022. SoftBart: Soft Bayesian Additive Regression Trees. arXiv preprint arXiv:2210.16375.

**Questions:**

1. I think it would be beneficial to provide more comparison between the proposed VaRT and conventional Bayesian decision tree models like BART and Bayesian CART, such as:

    1.1 One well-known advantage of BART is that it can perform variable selection to some extend. Does the proposed model have similar capability?

    1.2 What will be the computation benefits of the proposed method compared with the MCMC approach for conventional Bayesian decision tree models?

    1.3 In what scenario will the proposed VaRT outperform/underperform conventional Bayesian decision tree models?

2. Questions/comments on the experiments:

    2.1 Details of the experiments should be included. For example, is the RMSE in Table 1 computed on the test split? If so, how do you obtain the training/test splits from the data sets?

    2.2 In Table 1, I think it will be more fair to also compare with the 0.9 and 0.1 posterior percentiles of BART.

    2.3 Very limited discussion on the experiment results, especially for the UCI datasets.

3. Minor issues:

    3.1 In (3), is the denominator equal to 1? If so, the denominator can be trivially omitted.

    3.2 In Line 74, it is more rigorous to write "$s_k \sim \text{Bernoulli}(\gamma_k)$ independently for $k = 1, 2, \ldots$".

    3.3 Citation in Line 191 does not seem correct.


**Limitations:**

Some limitations and future work are discussed in the paper. I don't foresee any potential negative societal impact.

---

> ### Author Rebuttal · Authors · 2023-08-08
>
> We would like to express our gratitude for taking the time to review our manuscript. Your insightful comments and suggestions have been instrumental in guiding our work towards refinement, and we appreciate your thorough evaluation.
>
> Regarding your recommendation to evaluate our method against these models and other soft decision tree-based approaches, we want to clarify our approach. In the interest of providing a thorough evaluation within our current constraints, we chose to focus on established boosting methods like XGBoost, Random Forest, and CatBoost when conducting new experiments. The comparisons we've conducted in this rebuttal with widely recognized boosting methods offer insights into the strengths and performance of our approach. We believe these comparisons contribute valuably to our research and the broader field of Bayesian nonparametric regression.
>
> We fully understand your concern about the comparability of our method and acknowledge the challenges associated with implementing additional benchmarks and locating or implementing specific Python versions of certain models in such a short time-frame. Despite these challenges, we remain committed to addressing your valuable feedback in a meaningful way. In our revised manuscript, we will incorporate a detailed discussion of soft BART models and other relevant soft decision tree-based models. This will underscore the significance of these models in the context of Bayesian soft decision trees and contribute to a comprehensive understanding of the research landscape.
>
> Addressing your specific questions, we do believe our model possess the capacity for variable selection. We believe this could be achieved by incorporating global shrinkage priors into the splitting and prediction rules of VaRT. This clever observation was brought to our attention by reviewer Ya4w. We provide further clarification on this issue in the "global" rebuttal section but we remain open to answer any lingering questions.
>
> Reviewer 2ow5 suggested that we ran experiments on larger datasets. We took their advice and extended our evaluation to include datasets comprising more than 40,000 data points. Notably, our method demonstrated seamless scalability to handle these larger datasets, owing to its inherent parallelizability. The computational efficiency of our algorithm was evident, with fitting times spanning a range of one to three minutes. It's worth noting the significant contrast with Bayesian MCMC methods like BART, which are constrained by their sequential nature and lack support for parallelization. While we acknowledge the existence of rapid frequentist boosting algorithms such as XGBoost and CatBoost, we recognize the potential for further exploration in this direction. We are excited about the prospect of future advancements in this area and acknowledge that the initial implementation of our algorithm might not outpace these established algorithms in terms of speed.
>
> Moreover, we hope to have addressed your concerns on the experiments in the "global" rebuttal section. In particular, we have included a thorough discussion on our original and new experiments. Please feel free to reach out with any questions and are open to any feedback you may have. We intend to incorporate these discussions into our paper.
>
> Lastly, we appreciate your keen attention to detail, which led to the identification of the denominator's equivalence to 1 in equation (3). We have diligently revised our manuscript to reflect this clarification. Your suggestions in points 3.2 and 3.3 have also contributed to improving the clarity of our paper, and we have successfully incorporated these changes.
>
> Once again, we extend our heartfelt thanks for your invaluable feedback. Your expertise and insights have been instrumental in shaping the trajectory of our work, and we are dedicated to refining our manuscript in accordance with your suggestions.
>
> Respectfully,
>
> The Authors.

---

> > ### Comment · Reviewer_MwKc · 2023-08-16
> >
> > I thank the authors for the additional numerical results and their detailed response. The additional results show very competitive performance and demonstrate appealing computational benefits, which definitely strengthen the paper. I lean towards accepting this work.
> >
> > My conjecture is that the “soft” nature in the proposed method allows it to outperform conventional tree ensembles when the true mean function is relatively smooth. I encourage the authors to include comparisons with soft BART (open source R implementation available) and other tree ensemble methods with a larger number of trees (which can produce smoother prediction) in the revised version.

---

> > > ### Author Response · Authors · 2023-08-18
> > >
> > > Thank you for your thoughtful response and your engagement with the review process.
> > >
> > > Your observation regarding the potential advantage of our method in cases where the true mean function exhibits smoothness is quite astute. We found your conjecture intriguing, and your suggestion aligns well with the trends observed in our supplementary experiments. These initial findings are certainly encouraging for us to dive deeper into this hypothesis and explore its implications more comprehensively in our research.
> > >
> > > Once again, thank you for your expertise and thoughtful evaluation. Your insights have been incredibly helpful in strengthening the results of our paper.
> > >
> > > Best,
> > >
> > > The Authors

---

### Official Review · Reviewer_AUTX · 2023-07-07

**Soundness:** 3 good
**Presentation:** 3 good
**Contribution:** 3 good
**Rating:** 7
**Confidence:** 3

**Summary:**

The authors propose to use variational inference to train regression trees. The innovation lies in using variational inference as the optimization process rather than Markov Chain Monte Carlo. The authors demonstrate this method on 18 ML UCI problems and some causal inference and toy problems.

**Strengths:**

The paper is clearly written and specific about its contribution. The figures are generally good, although they could use better captions (especially Figure 1).

**Weaknesses:**

It's hard for me to judge the novelty and potential impact of this work. There are so many decision tree boosting algorithms out there that are blazingly fast (e.g. XGBoost, Catboost), whereas this method takes a few minutes to train. There could be an advantage if the resultant model was more interpretable than others, but that isn't really demonstrated. It is not clear to this reviewer whether BART is really a go-to, off the shelf ML algorithm like XGBoost has become, but the authors focus their side by side comparisons on this algorithm (which makes sense since it is perhaps the closest philosophically). In other words, the comparison would be stronger if a candidate boosted tree algorithm was included in the comparisons. MLPs are known to underperform most tree ensembles on UCI.

**Questions:**

The authors mention depth 5-7; what happens at smaller depths? A smaller tree would greatly aid interpretability.

**Limitations:**

there is no limitations section, although the authors discuss limitations.

---

> ### Author Rebuttal · Authors · 2023-08-08
>
> We sincerely appreciate the time and effort you've dedicated to reviewing our manuscript. Your thorough evaluation offers valuable insights that will undoubtedly enhance the quality and impact of this work.
>
> Regarding the concern on training time, we acknowledge that there are blazingly fast boosting algorithms available in the field. Your observation regarding the underperformance of MLPs on tabular datasets is also well-founded. To address these concerns and establish the feasibility and performance of our method across a diverse range of scenarios, we conducted a series of additional experiments. These experiments involved comparing the performance of our approach to well-established boosting algorithms, namely XGBoost, CatBoost, and Random Forest. We are excited to share that the performance of VaRT remained competitive and, in some cases, even outperformed these boosting algorithms across a diverse set of situations. We have included the results of these new experiments in Table A of the attached PDF for your reference and want to thank you for pointing out this gap in our evaluation.
>
> To showcase the scalability of our method, we also ran experiments on two new datasets containing more than 40,000 datapoints. Despite the increased data volume, the training time remained within the order of one to three minutes (see attached pdf on "global" rebuttal). This notable efficiency is attributed to the parallelizability of our approach, which offers a clear computational advantage over other Bayesian algorithms reliant on Markov Chain Monte Carlo.
>
> We hope that these additional experiments and insights provide a more comprehensive view of the potential impact of our work. Thank you once again for your valuable feedback and insights. Your suggestions have been instrumental in shaping our approach to further align with the expectations and demands of the field. We remain open to answering any other questions.
>
> Best,
>
> The Authors.

---

> > ### Comment · Reviewer_AUTX · 2023-08-14
> >
> > the additional comparisons to ensemble tree algorithms strengthens the paper. I encourage the authors to integrate this into the main text. i like the additional figure of model interpretation as well. I think this would should be accepted.

---

### Author Rebuttal · Authors · 2023-08-08

We extend our heartfelt gratitude to the reviewers for their insightful evaluations of our paper. The feedback we received has played a pivotal role in the evolution of our work, and we have approached our revisions with utmost seriousness and dedication.

One area that emerged as a significant opportunity for improvement was the lack of experimental comparison with frequentist tree-based boosting methods, a concern raised by reviewers AUTX, Ya4w, and MwKc. As emphasized by reviewer AUTX, our work draws philosophical inspiration from BART, which initially led us to benchmark our algorithm against it. However, we now recognize the necessity of providing a more inclusive comparison. In light of this, we have expanded our experiments to encompass three well-regarded boosting methods: Random Forests, XGBoost, and CatBoost. We are excited to share that even with the introduction of these additional benchmarks, our algorithm remains competitive with these state-of-the-art boosting methods. The results of these experiments are showcased in Table A of the attached pdf.

In the new experiments of Table A, we compared an ensemble of fifty trees of BART, Random Forests, XGBoost, and CatBoost to a single tree of our algorithm at depths of 3, 5, 7, and 10. Because of our commitment to transparency and rigorous evaluation, we have decided to report the results across all four runs of VaRT, as opposed to just reporting the best one. By adopting this approach, we aim to provide a thorough representation of VaRT's performance across various configurations, showcasing its consistency and versatility in diverse scenarios.

Our proposed method showcases notable strengths in comparison to established boosting methods as evidenced by the RMSE values presented in Table A. Across a diverse range of datasets, VaRT consistently demonstrates competitive performance, often outperforming or closely aligning with well-regarded boosting techniques such as CatBoost, RandomForest, and XGBoost. The versatility and effectiveness of VaRT's single-tree approach is evident from its ability to yield compelling results on various datasets, highlighting its potential to provide accurate predictions in different problem domains.

While our experiments were comprehensive, a couple of reviewers rightfully poinited out a lack of depth in our result discussions. To provide context, we employed a 90%/10% train-test split on 18 standard regression datasets from the UCI Machine Learning Repository. For fair comparison, features were standardized within (0,1) and (-1,1) ranges for Table 1 and Table A, respectively. The reported RMSE values in both tables are averaged from five test set runs, accompanied by their standard deviations.

Closer examination of Tables 1 and A reveals noticeable performance challenges for VaRT arise in datasets 'airfoil,' 'autos,' and 'sml.' This prompts us to scrutinize dataset attributes influencing VaRT's performance in these datasets. As we analyzed the experimental results, we observed variations in VaRT's performance across different datasets. While the majority of the datasets demonstrated strong performance, we noted that the 'autos' and 'airfoil' datasets posed unique challenges. Moreover it is worth pointing out that the 'sml' dataset constitutes a time-series task, which introduces complexities that may not align optimally with VaRT's current framework. These challenges have highlighted specific areas that require further investigation and improvement.

In comparison to other tree-based Bayesian algorithms, VaRT exhibits distinct advantages that enhance its practical utility. Notably, VaRT's framework is inherently parallelizable, a feature that sets it apart from MCMC-based methods like BART which are inherently sequential and lack this parallelization potential. This parallelizability not only accelerates the model training process but also empowers efficient exploration of large datasets, making VaRT particularly well-suited for modern computation demands. Furthermore, VaRT leverages gradient-based optimization techniques, making it compatible with widely used automatic differentiation engines such as PyTorch. We corroborated this by running new experiments on datasets with 40k+ datapoints where training took < 3 mins (see Table B on the attached pdf).

Reviewer Ya4w gave us with a couple of phenomenal suggestions that have greatly enriched our perspective on our work. One suggestion that particularly piqued our interest was the incorporation of cumulative shrinkage priors into the splitting and regression parameters. We believe this has the potential to significantly enhance the performance and interpretability of VaRT. As these priors are known for inducing sparsity in the posterior distribution, we believe that their addition could also address a question raised by reviewer MwKc on variable selection, as sparse solutions should lead to a model that uses only a subset of the features. We believe this to be a very promising and exciting avenue for future work.

Moreover, reviewer Ya4w's insight into viewing our model as a mixture model has been enlightening. We conducted experiments using simulated data from a mixture of Gaussians and found that the posterior distribution of our model indeed concentrates around the "true" clusters of the underlying data-generating model (see attached pdf). We hope our interpretation of this insight aligns with the reviewer's Ya4w intention, and we remain open to addressing any questions.

In conclusion, we extend our heartfelt appreciation to the reviewers for their invaluable insights and feedback that have significantly contributed to refining our paper. Your comments have been instrumental in shaping the trajectory of our research and enhancing its quality. In this regard, we will provide a detailed response to each of these points in the "individual" rebuttal section. We sincerely thank you for your time, expertise, and thoughtful guidance.

---

### Decision · Program_Chairs · 2023-09-21

**Decision:**

Accept (spotlight)

**Comment:**

This paper is in the same family of methods as BART, a very famous Bayesian cousin of the famous CART decision tree learning algorithms. Reasoning and working over the space of probability distributions over trees is not easy. The authors propose a variational inference based method to efficiently sample the posterior distribution, propose an efficient parallel implementation, and provide solid experimental comparisons. My main request is that, for the camera-ready version, the authors should include the additional experimental material included in their rebuttal.